# Mitochondrial Proteins as Source of Cancer Neoantigens

**DOI:** 10.3390/ijms23052627

**Published:** 2022-02-27

**Authors:** Gennaro Prota, Ana Victoria Lechuga-Vieco, Gennaro De Libero

**Affiliations:** 1Experimental Immunology, Department of Biomedicine, University Hospital Basel, University of Basel, Hebelstrasse 20, 4031 Basel, Switzerland; gennaro.delibero@unibas.ch; 2The Kennedy Institute of Rheumatology, Nuffield Department of Orthopaedics, Rheumatology and Musculoskeletal Sciences, University of Oxford, Oxford OX3 7FY, UK; ana.lechuga-vieco@kennedy.ox.ac.uk

**Keywords:** mitochondria, mtDNA mutations, T cell response, cancer neoantigens, post translational modifications

## Abstract

In the past decade, anti-tumour immune responses have been successfully exploited to improve the outcome of patients with different cancers. Significant progress has been made in taking advantage of different types of T cell functions for therapeutic purposes. Despite these achievements, only a subset of patients respond favorably to immunotherapy. Therefore, there is a need of novel approaches to improve the effector functions of immune cells and to recognize the major targets of anti-tumour immunity. A major hallmark of cancer is metabolic rewiring associated with switch of mitochondrial functions. These changes are a consequence of high energy demand and increased macromolecular synthesis in cancer cells. Such adaptations in tumour cells might generate novel targets of tumour therapy, including the generation of neoantigens. Here, we review the most recent advances in research on the immune response to mitochondrial proteins in different cellular conditions.

## 1. Genetics of Mammalian Mitochondria

Mitochondria are organelles with primary functions in producing energy and metabolic intermediates required for many cellular activities. They regulate several important cellular processes, including proliferation and differentiation, redox homeostasis, differentiation, and programmed cell death [1].

Recent novel phylogenomic analyses support the theory that mitochondria originated from an alphaproteobacterial endosymbiont [2,3]. During evolution, this endosymbiotic interaction gradually became further intertwined, with most mitochondrial genetic information being transferred to the nucleus. However, mitochondria retained vestigial DNA (mtDNA) with autonomous transcriptional and replicative machinery. Nuclear and mitochondrial genomes take advantage of a mutual regulatory system, which coordinates their respective functions through a series of signals, although these remain to be fully elucidated [4,5]. The mammalian mitochondrion comprises between 1000 and 1500 proteins, the vast majority of which are encoded by the nuclear genome, whilst a small number is encoded by the mitochondrial genome [6,7]. The human mitochondrial genome is a genetically compact, circular, double-stranded, 16.5 kb, large DNA molecule. Typically, between 100 and 10,000 mtDNA copies exist per cell, based on the specific cell type [8,9]. In mammalian cells, mtDNA is anchored to the inner mitochondrial membrane (IMM) within the mitochondrial matrix, packaged into protein–DNA complexes known as ‘nucleoids’, which are principally formed by the mitochondrial transcription factor A (TFAM) [10]. Human mtDNA encodes 11 mRNAs, 22 tRNAs, and 2 rRNAs [11]. Within the 11 mRNAs, 13 highly hydrophobic polypeptides are co-translationally inserted into the IMM, where they act as the core, membrane-bound subunits of the respiratory chain complexes I, III, and IV and ATP synthase. Moreover, mitochondrial proteins physically interact with other peptides produced by the nuclear genome to form respiratory chain complexes; alterations in the mtDNA sequence may therefore modify the assembly and function of each complex [12]. Such modifications may alter cell phenotype and functions, and consequently this paradigm is of particular interest in diseases, including cancer.

## 2. MtDNA Mutations and Cancer

The role of mtDNA mutations in cancer development and progression remains highly debated [13,14]. Historically, mitochondrial genetics in cancer was largely neglected, due to the attention paid to nuclear DNA but also to technical limitations [15]. A major difficulty in performing investigation stems from the fact that there is extensive heterogeneity in the mutational burden of the mitochondrial genome across multiple tumours [14]. Secondly, there are limited validated specific approaches which can be employed to determine the role of single nucleotide polymorphisms (SNPs) and mutations in mitochondrial genes. For example, it is difficult to knock out or silence mitochondrially-encoded genes, although recent studies have elegantly and initially addressed some of the technical hurdles [16]. This limitation can be overcome through employment of mitochondrial cybrids, cell lines in which endogenous mitochondria are substituted with mitochondria isolated from other cell types [17,18]. However, despite the apparent usefulness and elegance of this approach, it remains technically complex and requires testing for a large number of mutations, therefore limiting its use. This lack of information has resulted in several in silico studies which have attempted to correlate mutations and phenotypes [19,20].

It is common knowledge that mtDNA has a higher basal mutation rate compared with the nuclear genome [21]. Somatic mutations in the mitochondrial genome have been identified in studies comparing mtDNA sequences between cancer patients and matched controls in healthy and tumoral tissues. These studies showed that neoplastic cells accumulate mutations in mtDNA during cancer progression. The first evidence of mitochondrial somatic mutations was identified in 1998 in colorectal cancer cell lines [22], followed by studies in which mtDNA mutations were detected in solid and hematological cancers [23,24,25,26]. In one study, mtDNA mutations did not confer any advantage to tumorigenesis, suggesting that only random processes contributed to the homoplastic incidence of the mtDNA mutations [27]. More recently, neutral selection of missense mtDNA mutations was described, and no evidence of positive selection in truncating mutations was detected across 31 tumour types comparing 1675 mtDNA sequences from cancer and paired healthy tissues from patients [28]. It was later found that most mtDNA mutations in multiple tumours are due to replication errors and the selective pressure against deleterious mutations [29]. Finally, proof of selection against the transmission of deleterious mtDNA and propagation of functional mtDNA during oogenesis in the human germline was provided [30]. Such selection of intact mtDNA does not occur in cancer cells, in which mtDNA mutations instead accumulate and confer proliferative advantages [31]. Equally, somatic mtDNA mutations generate different mitochondrial-encoded respiratory complex peptides, evidenced in multiple tumour lineages [32]. Mutations in Complex I peptides increase both tumorigenesis and metastatic potential, directly linking consequences of mitochondrial genome alterations to cancer progression [33]. These findings were complemented with bioinformatic analysis showing elevated mutagenesis in genes encoding respiratory complex I subunits such as MT-ND1, MT-ND4, and MT-ND5 in kidney, thyroid, and colorectal cancers [31,32,34]. In conclusion, mutations in mitochondrial proteins are present in most cancer cells and correlate with modulations in tumour metabolic profiles and cancer cell metastatic potential. Despite these correlative studies associating mutations to tumour development, the same altered proteins could induce cancer-specific immune responses, with direct implications for the rational design of cancer vaccines.

## 3. Mitochondrial Mutated Proteins and Cancer Immunotherapy

T cell immunity against human cancer cells contributes to controlling tumour growth, and therapeutic enhancement of such immune responses has proven extremely effective in enhancing anti-tumour immune responses in select patients [35,36]. One major achievement has been the identification of immunogenic peptides present in tumour cells and not in healthy ones, thus allowing the induction of T cells specific for such peptides and capable of targeting only tumour cells. Methods of predicting cancer peptides representing neoantigens have been validated in a few cases, thus showing our current limited understanding of how cancer-specific peptides are generated and why their immunogenicity is limited [37]. These studies have not fully addressed the potential relationship between mtDNA mutations and the generation of tumour-specific neoantigens. Although mitochondria frequently develop mutations and other changes during cancer progression, very little is known about mitochondrial DNA-encoded antigens recognized by tumour-infiltrating cytotoxic T cells. Evidence was provided of a positive correlation between increased mitochondrial protein content and modulation of responses to immunotherapy in melanoma patients [38]. These effects were attributed to an increase in the efficiency of peptide presentation by MHC molecules and to type I and II interferon signaling [38]. Analysis of patient melanoma samples showed that mutations in the mitochondrial-encoded respiratory complex III subunit cytochrome b gene (Cytb) induce peptide sequence modifications leading to neopeptides stimulating specific CD4+ T cells. Most Cytb mutations were T→C and A→G, and this protein product encoded by a mitochondrial gene mutation was present in the cytoplasm and in late endosomal compartments, permitting loading onto MHC class I and II molecules, respectively [39]. The exact mechanism of how these proteins cross membranes remains unclear. These studies provided the first convincing evidence that mutated peptides encoded in mitochondrial genome may induce specific immunity.

In the same manner, a cancer vaccine was formulated using an enriched mitochondrial protein extract from cancer cells. Prophylactic injections were followed by tumour challenge, resulting in delayed tumour growth in the vaccinated group compared to the control group. Evidence was also provided that mutations in mitochondrial proteins generated neo-antigens inducing the tumour-specific immune response [40]. Thus, this study convincingly showed the therapeutic potential of T cells specific for mitochondrial neoantigens.

Our research contribution built upon these results and showed that mitochondrial location of T cell-stimulatory proteins significantly enhances priming of antigen-specific CD8 T cells in vivo [41]. In the same study, mitochondria purified from B16 melanoma cells injected as vaccines induced potent CD8 T cell responses that protected mice challenged with the same tumour cells. The immunogenicity of mitochondrial proteins was also confirmed in a cancer patient. By comparing the sequences obtained from tumour and germline DNA, tumour-specific, nonsynonymous single-nucleotide variations were identified and 60 neoepitopes were predicted. Using HLA tetramers loaded with mitochondrial mutated peptides, CD8 T cells specific for 4 neoantigens derived from 2 mitochondrial-localized mutated proteins were identified [41].

Several studies confirmed that mitochondrial localization of proteins correlates with their immunogenicity. Proteins such as aconitase [42], the MLRQ subunit of the NADH ubiquinone oxidoreductase [43], and the E2 component of the pyruvate dehydrogenase complex [44,45], which are encoded in the nucleus but resident in mitochondria, are highly immunogenic. The mechanism whereby mitochondrial localization confers increased immunogenicity is poorly defined. Cross-priming events, which are facilitated by both the frequency of mitochondria-localized proteins and the activation of the STING–cGAS pathway, probably play an important role [41]. Indeed, immunogenicity of the prototype antigens NY-ESO-1 and OVA is higher when they are localized in mitochondria than in cytoplasm of tumour cells [41]. This promising field of investigation requires additional studies to understand the molecular mechanisms linking mitochondrial localization to increased immunogenicity, to address the question of how frequently T cell responses are directed against peptides encoded by mutated mtDNA and to evaluate these immune cells in patients with different types of cancers.

## 4. Post-Translational Modifications: A Different Way to Generate Neoepitopes?

Post translational modifications (PTMs) play a key role in numerous biological processes by significantly affecting the structure and function of proteins [46]. PTMs contribute to several alterations in cancer cells including the regulation of tumour suppressor proteins or oncogene activity [47,48]. Unique PTMs may also generate neoepitopes [49,50], thus harnessing the scientific interest for using modified proteins as new anti-cancer vaccines [51,52]. Amongst PTMs, peptide oxidation [53], phosphorylation [54,55], and citrullination [56] have the potential to generate unique pools of antigens shared by cells with similar metabolic alterations. Post-translational modifications of proteins may also impact mitochondrial dynamics, protein quality control, and, ultimately, mitochondrial performance [57]. SUMOylation, S-oxidation (sulfinylation and sulfenylation), S-glutathionylation, S-nitrosylation, acetylation, phosphorylation, glycosylation, protein cleavage, crotonylation, ubiquitination, and ADP-ribosylation are PTMs reported in the mitoproteome [57,58]. All these PTMs generate new antigens independently of their encoding genes, and in several instances these modifications promote MHC binding and TCR/MHC interactions.

Below, we briefly summarize most known PTMs with their effects on T cell recognition.

### 4.1. Oxidation

Reactive oxygen species (ROS) are signaling molecules involved in several physiological processes, including the regulation of mitochondrial homeostasis, cell differentiation, immune responses, and ageing. In the cell, ROS are generated via different routes, which may involve cytochrome p450, peroxisomes, xanthine oxidase, and NADPH oxidase. However, in non-phagocytic cells, the main sources of ROS are respiratory complexes I and III in the mitochondrial electron transport chain [59]. ROS generation can be important in antigen-specific T cell expansion [60]. ROS may also affect the oxidation state of peptide antigens, thus inducing the appearance of slightly different peptides that may show increased interaction with specific TCRs [53]. Accordingly, cancer-derived oxidized peptides can stimulate specific T cells recognizing ovarian epithelial cancer cells [61]. Whether oxidation of mitochondrial proteins may also generate cancer-specific immunity remains to be investigated.

Hydrogen peroxide (H_2_O_2_) promotes PTM of peptides through oxidation of cysteine residues, with subsequent effects on cell signaling. Under physiological pH conditions, cysteine residues exist as thiolate anions (Cys-S-), and in the presence of H_2_O_2_ the thiolate moiety is oxidized to its sulfonic form (Cys-SOH). Such oxidation process is reversible, with thioredoxins and glutaredoxins being involved in protein reduction. Oxidation of a Cytomegalovirus (CMV) immuno-dominant peptide decreased its antigenicity by hampering its interaction with the TCR, thus implying that during infection the oxidative state of infected cells may affect recognition by virus-specific T cells [53]. On the other hand, evidence of increased peptide immunogenicity after oxidation is still lacking.

### 4.2. Phosphorylation

Phosphorylation of mitochondrial peptides is a rapid and reversible modification which influences many important cellular activities, including metabolic signaling. More than 90% of the mitochondrial proteins contain at least one phosphorylation site, with an average of 8 phosphorylation sites in all annotated, mitochondrial-located proteins [62]. Phosphorylation modifications compete with glycosylation in serine/threonine residues, and therefore, they may both affect the immunogenicity of peptides carrying such modifications. One example is the chaperone Hsp60, located in the mitochondrial compartment. PTMs through glycosylation, phosphorylation, and hyperacetylation can alter the functional domains of this protein [63]. Some of these modifications may induce conformational changes sufficient to enhance the recognition by specific TCRs [49]. The immunogenicity of peptides abnormally phosphorylated in cancer cells is an active field and different screening approaches have been established to identify MHC I- and MHC II-binding tumour-specific phosphopeptides [49,50,54]. Clinical trials evaluating the therapeutic use of phosphorylated peptides as cancer vaccine are ongoing [64,65].

## 5. Mechanism of Mitochondrial Antigen Direct and Cross-Presentation in Cancer

In most cases, tumour cells do not prime specific immune responses, probably as a result of low co-stimulation. Conversely, dendritic cells (DCs) behave as professional antigen-presenting cells, capable of efficient internalization of tumour debris and IL-12-production [66,67]. DCs may also take exogenous proteins and prime MHC class I-restricted T cells, a function identified as cross-priming. With this mechanism they may efficiently prime tumour-specific killer T cells [66,67]. DC may take up mitochondrial proteins within the tumour microenvironment, migrate to draining lymph nodes, and cross-present tumour antigens to MHC class I-restricted T cells [41]. The mechanism leading to mitochondria release by tumour cells and their uptake by DC is not understood. The transfer of mitochondria or mitochondrial proteins might involve tunnelling nanotubes (TNTs) [68,69], extracellular vesicle (EVs) uptake [70], and cytoplasts uptake [71] (Figure 1). Recent findings showed that a mechanism of quality control of mitochondrial integrity and function is represented by a release of free naked mitochondria [72]. The uptake of such released damaged organelles by professional antigen-presenting cells might represent an additional mechanism promoting presentation of mitochondrial proteins. A recent study also outlined that on some occasions an exchange of mitochondria between cells may occur [73]. Whether this organelle trafficking occurs in diseases and affects presentation of mitochondrial antigens remains unknown.

Another important aspect is how mitochondrial proteins are handled by tumour cells before their processing and binding to MHC molecules. Protein immunogenicity is determined via localization within a peculiar cellular compartment. Indeed, when the same protein is localized in the cytoplasm or in mitochondria, different peptides are generated and stimulate T cells with different specificities [74]. Thus, mitochondrial localization determines the type of processing.

Autophagy allows recycling of self-components through lysosomal degradation and is involved in the presentation of endogenous antigens by both MHC class I and class II molecules [75]. Accordingly, mitophagy, a selective process of autophagy that selectively degrades damaged mitochondria, might play a role in the presentation of mitochondrial antigens, although few data indicate such possibility [76].

A mechanism relying on the generation and trafficking of mitochondria-derived vesicles (MDVs) rather than on autophagy/mitophagy was described in a model of Parkinson’s disease [77]. These studies showed that stress conditions such as heat stress or LPS stimulation induced the formation of MDV in a monocyte/macrophage-like cell line. The formed vesicles could fuse with late endosomes/lysosomes where mitochondrial antigens are processed by hydrolytic enzymes. The authors speculated that mitochondrial peptides are released in the cytosol, where they are processed by the proteasome, allowing their translocation in the endoplasmic reticulum and presentation by MHC class I molecules to T cells [77,78]. In normal cells, this process is constitutively suppressed by Parkin, a known regulator of mitophagy that is frequently defective in Parkinson’s disease patients [79]. Recent evidence indicates that defects in the mitophagy machinery may also have a role in the progression of cancer. Indeed, in several cancer types Parkin function is lost [80]. The absence of Parkin in tumour cells might permit the accumulation of MDVs and thus induce mitochondrial antigen presentation with a mechanism similar to that observed in Parkinson’s disease. It will be of interest to investigate whether other kinds of stress (i.e., hypoxia, proliferative stress, lack of nutrients, cytokine stimulation) may induce MDV formation thus promoting presentation of mitochondrial antigens in cancer cells.

## 6. Immune Response to Mitochondrial Proteins in Diseases Other Than Cancer

Several studies showed that mitochondrial proteins may induce specific immune responses. Immune recognition of mitochondrial-encoded proteins was first reported by Kelley et al. in 1974, who observed that myocardial infarction in dogs triggers auto-antibodies against antigens located in the outer and inner mitochondrial membrane [81]. Later, it was observed that patients affected by autoimmune diseases may generate T cell responses against mitochondrial mutated self-peptides and that in some instances these T cells cross-reacted with the same non-mutated proteins [82]. On the contrary, in healthy donors T cells recognizing mitochondrial mutated self-proteins did not cross-react with non-mutated ones, suggesting that autoimmune responses can arise upon expansion of T cells recognizing both mutated and wild-type mitochondrial proteins. In addition, mouse studies showed that mitochondrial proteins may induce strong immune responses. In some mouse strains, skin graft rejection was dependent on unknown factors inherited from the mother. Maternal inheritance of these characteristics suggested a possible implication of a gene present in mtDNA [83]. This antigen was later identified as the amino-terminal peptide of ND-1, a protein encoded within mtDNA, that is presented by H2-M3, an MHC class I-like molecule [83,84]. As the H2-M3 gene is not present in the human genome, there is no homologous system presenting this peptide to human T cells.

In a few studies, mtDNA-encoded proteins were investigated for their possible relevance in transplant rejection. A landmark study showed that the rejection of autologous, induced pluripotent stem cells (iPSC) can be mediated by T cells specific for mutated proteins encoded by mtDNA. Consequently, in vitro expanded iPSCs accumulate de novo mtDNA mutations encoding altered self-peptides presented by select MHC alleles in humans and mice [85].

## 7. Conclusions

The ability of the human immune system to recognize and mount an immune response against mitochondrial mutated proteins has been observed under different pathological conditions. Mutations in mtDNA have been observed in several cancers, and whether such mutations induce specific immunity is a topic of active investigation. Several questions remain open: (i) what is the relevance of cross-presentation of mitochondrial antigens, in relation to cellular components exchange between DCs and tumour cells; (ii) how are mitochondrial proteins handled by tumour cells before their processing and loading on MHC molecules; (iii) do mtDNA mutations have a role in specific recognition of cancer cells by T lymphocytes; (iv) which mutated peptides become immunogenic, considering their cellular localization and post-translational modifications; (v) how frequently mitochondrial PTMs generate neoantigens in cancer cells. All these issues are of great interest and will reveal novel features of the physiological role of mitochondria in the immune response.

## Figures and Tables

**Figure 1 ijms-23-02627-f001:**
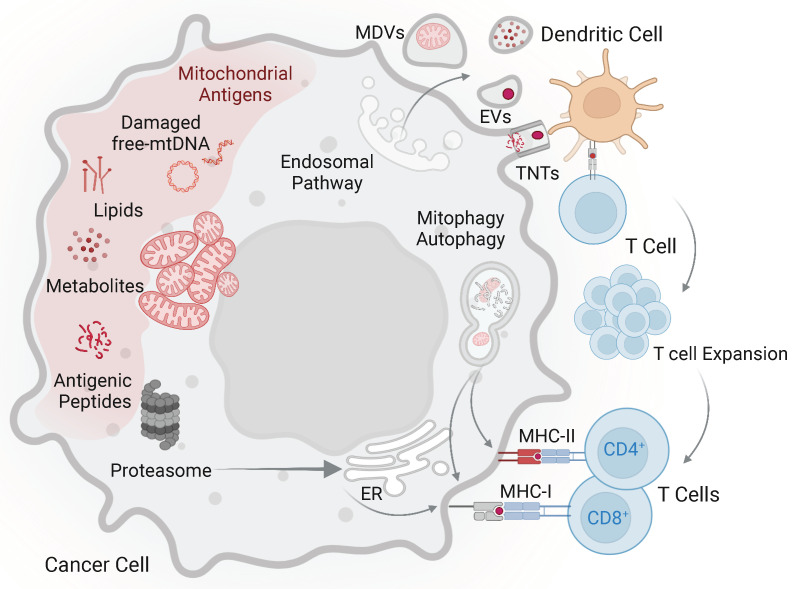
Mitochondrial-derived antigens and possible mechanisms promoting their immunogenicity. In tumour cells neo-antigens can be generated via mutations of mitochondrial DNA and different modifications compared to normal cells. The uptake of mitochondria and mitochondrial debris by dendritic cells may involve tunnelling nanotubes (TNTs), extracellular vesicles (EVs), and mitochondria-derived vesicles (MDVs). Tumour antigens (red dot) are presented to specific T cells, which in turn expand and acquire the capacity to recognize tumour cells. Mitochondrial proteins released in the cytoplasm may be processed by proteasome, translocated into the endoplasmic reticulum, and presented by MHC class I molecules. Autophagy is involved in the presentation of endogenous antigens by both MHC class I and class II molecules. Accordingly, mitophagy, a form of macro autophagy that selectively degrades damaged mitochondria, might play a role in presentation of mitochondrial antigens. Other components of mitochondria from tumour cells, including lipids and metabolites, could also represent antigens stimulating specific T cells. However, experimental evidence is still lacking.

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
