# Peer review of "Mitochondrial Proteins as Source of Cancer Neoantigens"

_ijms, 2022, doi:10.3390/ijms23052627_

Round 1
Reviewer 1 Report
Interesting the study by Prota et al. on mutations in mitochondrial proteins present in most cancer cells and correlatable with modulations in tumor metabolic profiles and cancer cell metastatic potential. These are mitochondrial changes resulting from the high energy demand and the increase in macromolecular synthesis in tumor cells: they are adaptations that generate new targets of tumor therapy, including the generation of neoantigens.
However, there are so many unsolved questions to understand the molecular mechanisms linking mitochondrial localization to increased immunogenicity.
I'm not thrilled about it. I would say it is a study without infamy and without praise.
Chapter 4 "Immune response to mitochondrial proteins in diseases other than cancer" creates confusion placed in the midst of 'cancer'. Among other things, it does not add salient information to the main topic of the entire study. Therefore, I would consider it appropriate to delete it or move it from there.
The final revision and reading of the manuscript - carried out before submission - leave much to be desired: there are inaccuracies of editing:
• What are DCs? (line 242)
• Points 4. and 5. are repeated twice, therefore the numbering of the chapters must be redone.
• At the end of the sentence “Below, we briefly summarize most known PTMs with their effects on T cell recognition:”, the colons need to be replaced with a full stop.
Author Response
Interesting the study by Prota et al. on mutations in mitochondrial proteins present in most cancer cells and correlatable with modulations in tumor metabolic profiles and cancer cell metastatic potential. These are mitochondrial changes resulting from the high energy demand and the increase in macromolecular synthesis in tumor cells: they are adaptations that generate new targets of tumor therapy, including the generation of neoantigens. However, there are so many unsolved questions to understand the molecular mechanisms linking mitochondrial localization to increased immunogenicity.
I'm not thrilled about it. I would say it is a study without infamy and without praise.
We thank the reviewer for the comments.
Chapter 4 "Immune response to mitochondrial proteins in diseases other than cancer" creates confusion placed in the midst of 'cancer'. Among other things, it does not add salient information to the main topic of the entire study. Therefore, I would consider it appropriate to delete it or move it from there.
Regarding chapter 4 we think it reinforces the concept that mitochondrial proteins can be immunogenic, listing a series of studies that are not strictly related to cancer. We agree with the reviewer that the position of such paragraph slows down the flow of the reading, so we moved it at the bottom.
The final revision and reading of the manuscript - carried out before submission - leave much to be desired: there are inaccuracies of editing:
- What are DCs? (line 242)
We thank the reviewer for this observation. DCs is referring to Dendritic Cells. We have added this information in the modified manuscript.
- Points 4. and 5. are repeated twice, therefore the numbering of the chapters must be redone.
We thank the reviewer for pointing this out and we have included the requested changes in the revised manuscript.
- At the end of the sentence “Below, we briefly summarize most known PTMs with their effects on T cell recognition:”, the colons need to be replaced with a full stop.
We thank the reviewer for pointing this out and we have included the requested changes in the revised manuscript.
Reviewer 2 Report
In this review Prota et al. discuss the most recent advances on the immune response to mitochondrial proteins in different cellular conditions.
The manuscript is well written and presents some unexplored aspects related to physiological role of mitochondria in the immune response.
However, in my opinion, Authors should better clarify the link between metabolic rewiring, the switch of mitochondrial functions and the immune response against mitochondrial mutated proteins.
Author Response
In this review Prota et al. discuss the most recent advances on the immune response to mitochondrial proteins in different cellular conditions.
The manuscript is well written and presents some unexplored aspects related to physiological role of mitochondria in the immune response.
We warmly thank the reviewer for the review and comments about our work.
However, in my opinion, Authors should better clarify the link between metabolic rewiring, the switch of mitochondrial functions and the immune response against mitochondrial mutated proteins.
The reviewer has raised a very interesting point of discussion.
Cancer cells increase metabolic rate under a hypoxic microenvironment for faster proliferation. This overall metabolic rewiring increases the reactive oxygen species (ROS) production, one of the hallmarks of cancer. ROS is the main source of mtDNA damage, including single-strains breaks, and nucleotide oxidations. Mitochondrial mutation accumulation has been described in different types of cancer with negatives consequences in OXPHOS enzymes and facilitating the adaptation of the cancer cell to demanding metabolic requirements in the tumor microenvironment. The increased number of mutated-self peptides have been also shown to increase the expression of MHC class I, facilitating the presentation of antigens to immune cells [1] and triggering autoimmunity [2]. Based on this idea, Pierini et al. designed one of the first vaccines based on mtDNA-mutated peptides and showed constrained tumor growth by T cell-mediated immunity [3]. Although cells can survive with up to 80% of mutant mtDNA [4], and increased mitochondrial removal is observed in cells with elevated mutant load [5]. Damaged mitochondria can be degraded by i) autophagy, which triggers MHC class I and II antigen presentation [6]; and, ii) selective removal of mitochondria independent of PINK1-Parkin that triggers the mitochondrial-derived vesicles (MDVs) trafficking and presentation of the antigenic peptides by the MHC class I complex [5]. Therefore, mitochondrial function in the cancer cell is linked to immune surveillance. We have tried to summarize these concepts in the manuscript, but it’s true that the literature falls short in the connection between metabolic rewiring and immune response against mitochondrial mutated proteins.
Like the reviewer, we have strong interests in understanding how changes in mitochondrial metabolism can alter immune responses and we agree that addressing this point would be of great interest to the scientific community. We are submitting a manuscript in which we engineered transmitochondrial cybrids from B16F10 melanoma cells (identical nDNA, different mtDNA variants). This tool allowed us to modulate the mitochondrial metabolism and the mitoproteome of cancer cells to examine whether the metabolic changes can qualitatively and quantitatively alter epitopes and whether knowledge of these changes can be used to identify more effective tumour specific epitopes for targeting by the immune system. We characterized the modifications in the peptidome to determine novel neo-epitopes and found differences in the efficiency of the T cell immune response that is primed by such tumours. Our results show that substitution of different mtDNA variants is sufficient to promote differences in mitochondrial function, metabolic exhaustion in tumor infiltrating T lymphocytes and a concomitant cellular adaptive response. Therefore, these investigations will form the basis of future studies which specifically addresses how different amino acidic sequence in mitochondrial encoded peptides promotes metabolic changes and interactions with the immune cells in the tumor microenvironment.
- Gu, Y., et al., Role of MHC class I in immune surveillance of mitochondrial DNA integrity. J Immunol, 2003. 170(7): p. 3603-7.
- Chen, L., et al., Experimental evidence that mutated-self peptides derived from mitochondrial DNA somatic mutations have the potential to trigger autoimmunity. Hum Immunol, 2014. 75(8): p. 873-9.
- Pierini, S., et al., A Tumor Mitochondria Vaccine Protects against Experimental Renal Cell Carcinoma. J Immunol, 2015. 195(8): p. 4020-7.
- Rossignol, R., et al., Mitochondrial threshold effects. Biochem J, 2003. 370(Pt 3): p. 751-62.
- Matheoud, D., et al., Parkinson's Disease-Related Proteins PINK1 and Parkin Repress Mitochondrial Antigen Presentation. Cell, 2016. 166(2): p. 314-327.
- Munz, C., Autophagy proteins in antigen processing for presentation on MHC molecules. Immunol Rev, 2016. 272(1): p. 17-27.